# Enhanced propagation of motile bacteria on surfaces due to forward scattering

Stanislaw Makarchuk[1], Vasco C. Braz[2,3], Nuno A.M. Araújo[2,3], Lena Ciric[4] & Giorgio Volpe [1]

How motile bacteria move near a surface is a problem of fundamental biophysical interest and is key to the emergence of several phenomena of biological, ecological and medical relevance, including biofilm formation. Solid boundaries can strongly influence a cell's propulsion mechanism, thus leading many flagellated bacteria to describe long circular trajectories stably entrapped by the surface. Experimental studies on near-surface bacterial motility have, however, neglected the fact that real environments have typical microstructures varying on the scale of the cells' motion. Here, we show that micro-obstacles influence the propagation of peritrichously flagellated bacteria on a flat surface in a non-monotonic way. Instead of hindering it, an optimal, relatively low obstacle density can significantly enhance cells' propagation on surfaces due to individual forward-scattering events. This finding provides insight on the emerging dynamics of chiral active matter in complex environments and inspires possible routes to control microbial ecology in natural habitats.

[1] Department of Chemistry, University College London, 20 Gordon Street, London WC1H 0AJ, UK. [2] Centro de Física Teórica e Computacional, Faculdade de Ciências, Universidade de Lisboa, P-1749-016 Lisboa, Portugal. [3] Departamento de Física, Faculdade de Ciências, Universidade de Lisboa, P-1749-016 Lisboa, Portugal. [4] Department of Civil, Environmental and Geomatic Engineering, University College London, Gower Street, London WC1E 6BT, UK. Correspondence and requests for materials should be addressed to G.V. (email: g.volpe@ucl.ac.uk)

Microorganisms live in natural environments that present, to different extents, physical, chemical and biological complexity[1,2]. This heterogeneity influences all aspects of microbial life and ecology in a wide range of habitats, from marine ecosystems[3] to biological hosts[4]. For example, flow and surface topology can trigger or disrupt quorum sensing in bacterial communities[5–7] as can shape dynamics of microbial competition in biofilms[8]. To enhance their fitness within such complexity, several bacterial species, e.g. *Escherichia coli* bacteria[9], are motile, which is key in promoting many biologically relevant processes, such as the formation of colonies and biofilms on surfaces[1,2,10]. Justified by fundamental biophysical curiosity as well as by the ecological and medical relevance of biofilms[11–13], significant research effort has, therefore, been devoted to elucidate the dynamics of bacterial near-surface swimming. We now know that, due to hydrodynamic interactions[14–16], several flagellated bacteria tend to describe circular trajectories when swimming near surfaces[13,17–22]. The interaction with a physical boundary can also lead to escape times that are much longer than the typical reorientation times for bulk swimming[23–26], thus resulting in long stable trajectories on surfaces that can eventually promote cell adhesion[14,27–30]. Surprisingly, even though natural bacterial habitats present characteristic features that vary on a spatial scale comparable to that of the cells' motion[7,8], experimental studies of near-surface swimming have mainly focused on smooth surfaces devoid of this natural complexity. Nonetheless, for far-from-equilibrium self-propelling particles, such as motile bacteria, both individual and collective motion dynamics can depend on environmental factors in non-intuitive ways, as recently shown for microscopic non-chiral active particles numerically[31–35] and experimentally[36,37]. Moreover, in environments densely packed with periodic patterns of obstacles, turning angle distributions of bacterial cells change from bulk swimming and their trajectories can be efficiently guided along open channels in the lattice[38,39].

Here we show that the motion of individual *E. coli* cells swimming near a flat surface is strongly influenced by the presence of micro-obstacles of size comparable to the typical bacterial cell. Counterintuitively, at low obstacle densities, the peritrichously flagellated bacterial cells diffuse ≈50% more efficiently than on a smooth surface. The interaction with the obstacles can, in fact, rectify the cells' near-surface motion chirality over distances orders-of-magnitude longer than the typical cell size. This behaviour is fundamentally different from that of non-chiral active colloids cruising through random obstacles with a fixed motion strategy, which instead get more localised for increasing obstacle densities[31,40,41]. For chiral bacteria, the expected behaviour is only observed at higher densities, consistently with previous observations of *E. coli* cells swimming in quasi-two-dimensional (2D) porous media[42]. We develop, and verify numerically, a microscopic understanding of the transition between enhanced surface propagation and localisation by identifying two types of cell–obstacle interactions, namely forward-scattering events and head-on tumble-collisions.

## Results

**Near-surface swimming with micro-obstacles.** To identify how the spatial heterogeneity on flat surfaces influences the propagation of bacteria, we recorded trajectories of motile *E. coli* cells swimming near a glass surface in a quasi-2D geometry with different densities $\rho$ (defined as fractional surface coverage) of fixed obstacles in the range $0\% \leq \rho \leq 12\%$ (Methods). *E. coli* bacteria are peritrichously flagellated prokaryotic cells that swim through an alternation of run and tumble events[9]. Consistent with previously reported sizes after cell division[9], the typical bacterial cell in our experiments was $2.6 \pm 0.7$ μm long and

$1.2 \pm 0.4$ μm wide (estimated from microscopy images). When swimming near a smooth surface, *E. coli* cells move in long circular trajectories[14,17,18], which are typically stably entrapped by the surface[14,27,28,30,43]. We estimated the average translational and angular speeds of the motile cells in our experiments to be $\langle v \rangle = 11 \pm 4$ μm s$^{-1}$ and $\langle \Omega \rangle = 0.8 \pm 0.5$ rad s$^{-1}$, respectively (Supplementary Fig. 1 and Methods). The 10-s-long trajectories in Fig. 1a, along with Supplementary Fig. 1b, highlight the experimental spread in $\Omega$, which spans from 0 rad s$^{-1}$ (non-chiral) to 2.5 rad s$^{-1}$ (strongly chiral), due to both intercell variability and distance variations of the cells from the two surfaces of the sample chamber.

When the bacterial cells swim near a surface with a complex microstructure as in Fig. 1b, interactions with the fixed obstacles become unavoidable. These interactions can significantly affect a cell's propagation over the surface. For example, the trajectory in Fig. 1b frequently slows down or stops near the obstacles, which can sterically impede the cell's progression until its direction of motion changes to point away from them. To quantify the influence of these interactions on the cells' motion as a function of $\rho$, we considered how efficiently the bacteria can propagate through a circular area of radius $R$ (Fig. 1b and Methods). We initially set $R = 25$ μm, i.e. one order of magnitude longer than the typical cell's length. For all cells that propagate through any such area at a given $\rho$, we can assign an average effective propagation distance $L_{\mathrm{eff}} \in [0, 2R]$ as a function of the obstacle density (Fig. 1c and Methods). This quantity measures the average distance run by the cells when crossing the circular area rather than their average path length[44]: independently of the actual path taken by each trajectory within the corresponding area, the two limit values of $L_{\mathrm{eff}}$ respectively represent the cases where all cells exit from where they entered or at the diametrically opposite point. Figure 1c shows that, without obstacles ($\rho = 0\%$), $L_{\mathrm{eff}} \approx R$. This value has a purely geometrical meaning as it closely corresponds to the length (≈24 μm) of the common chord at the intersection between the circular area and the average circular trajectory (with radius $R_{\mathrm{EC}} = \frac{\langle v \rangle}{\langle \Omega \rangle} = 13.7$ μm) of the *E. coli* cells propagating within it when entering perpendicularly to the area perimeter. Counterintuitively, instead of hindering propagation as for non-chiral active particles[41], a slight increase in $\rho$ ($2\% \leq \rho \leq 8\%$) allows bacterial cells to propagate over longer distances than on a smooth surface (with an ≈20% peak enhancement at $\rho = 2\%$). The more intuitive behaviour, where $L_{\mathrm{eff}}$ decreases for increasing $\rho$, is only observed at higher obstacle densities ($\rho > 8\%$).

The previous result suggests that a few micro-obstacles have a beneficial effect on the capability of chiral bacteria to swim over large distances near surfaces, and only become detrimental at high densities. To account for differences in the time spent by the bacteria within an area for different obstacle densities, we also calculated the cells' normalised average effective speed $V_{\mathrm{eff}}$ as a function of $\rho$ (Fig. 1d and Methods). This quantity shows a similar trend to $L_{\mathrm{eff}}$. Initially, for $2\% \leq \rho \leq 4\%$, the cells propagate faster than on a smooth surface due to the increase in $L_{\mathrm{eff}}$ (with an ≈12% peak enhancement at $\rho = 2\%$). However, unlike $L_{\mathrm{eff}}$, $V_{\mathrm{eff}}$ at $\rho = 6\%$ is already comparable with the value at $\rho = 0\%$ and rapidly decreases thereafter, as more frequent encounters with the obstacles increasingly prolong the cells' residence time within the area. These variations in $V_{\mathrm{eff}}$ with $\rho$ are also reflected in the spatial distribution of the cells on the surface (Fig. 1e): while at low obstacle densities ($\rho = 2\%$) this distribution is basically uniform in space as for $\rho = 0\%$, it becomes more heterogenous at higher obstacle densities, as localisation hot spots start to emerge in the proximity of the obstacles.

By analysing typical trajectories (Fig. 2a–e), we can qualitatively appreciate how cell–obstacle interactions are directly

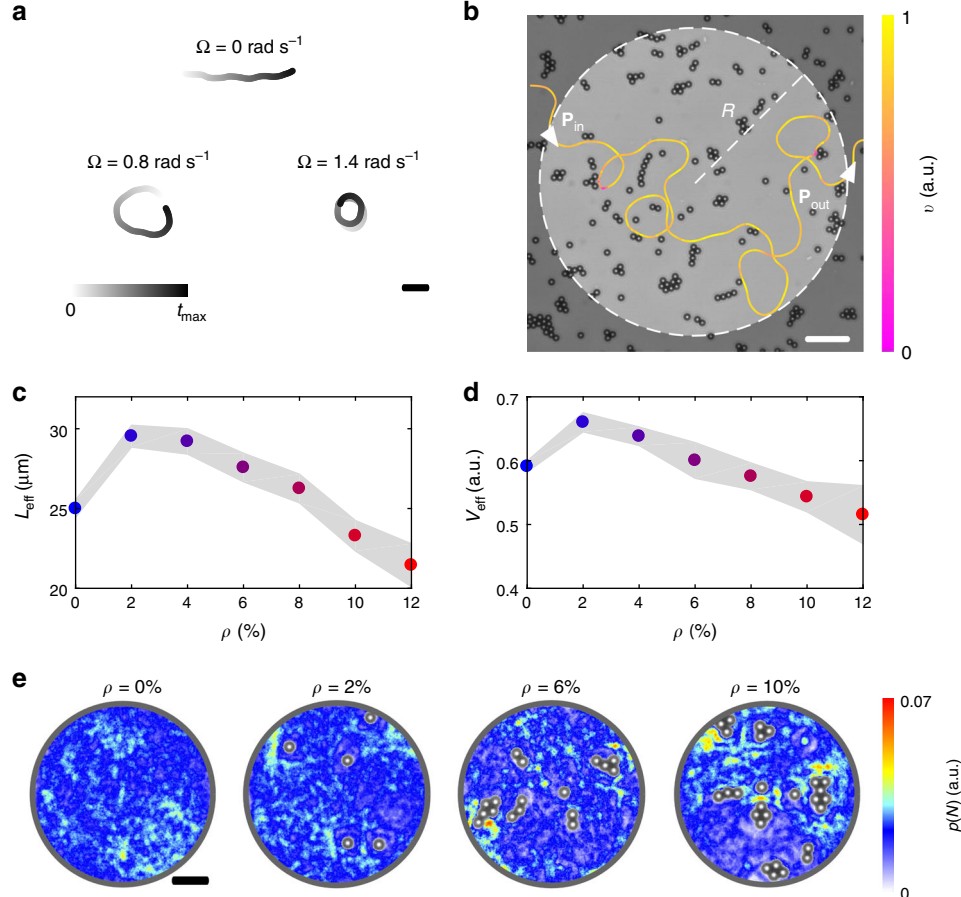

**Fig. 1** Propagation and localisation of *E. coli* cells near surfaces with micro-obstacles. **a** Exemplary 10-s-long trajectories of *E. coli* cells swimming near a surface in the absence of obstacles ($\rho = 0\%$) for different angular speeds $\Omega$. The case for $\Omega = 0.8$ rad s$^{-1}$ corresponds to the average value of angular speed in our experiments. The shading represents the trajectory's time evolution. The black scale bar corresponds to 20 μm. **b** Exemplary trajectory of an *E. coli* cell swimming near a surface with fixed obstacles. The trajectory's colour code represents the cell's instantaneous velocity $v$ normalised to its maximum value. The white dashed line delimits a circular area of radius $R$ in the field of view and intersects the trajectory at points $\mathbf{P}_{in}$ and $\mathbf{P}_{out}$, which respectively represent the cell's points of entrance and exit in the circular area. This geometrical configuration is used for the calculation of the average effective propagation distance $L_{eff}$ in **c** and normalised speed $V_{eff}$ in **d** (Methods). The white scale bar corresponds to 20 μm. **c, d** Average effective propagation distance $L_{eff}$ and normalised speed $V_{eff}$ as a function of the obstacle density $\rho$ for a circular area of radius $R = 25$ μm. Each value is obtained from averaging over at least 1000 different trajectories. The shaded area around the average values represents one standard deviation. The values of obstacle density $\rho \geq 2\%$ are given with a 0.6% standard deviation. **e** Spatial probability density maps $p(N)$ of finding individual bacterial cells within a circular area of radius $R = 25$ μm for increasing obstacle densities $\rho$ over 1-h-long experiments. Each map was calculated from at least 450 different trajectories and an occupied pixel was only accounted for once for each trajectory. The black scale bar corresponds to 10 μm

responsible for the observed trends in $L_{eff}$ and $V_{eff}$. As shown by the probability distributions of the change in effective propagation direction $\Delta\theta_{eff}$ (Fig. 2a–e and Methods) and by the trajectories in Supplementary Fig. 2, all propagation behaviours are possible at any $\rho$. However, these distributions are not necessarily uniform: different propagation directions are indeed favoured at different $\rho$ values, as shown by the average change in effective propagation direction $\Delta\Theta_{eff} = \langle\Delta\theta_{eff}\rangle$ (Fig. 2f and Methods). Without obstacles (Fig. 2a), the circular near-surface swimming of the bacteria typically induces a u-turn, thus making them exit near their entrance point. Due to the chirality in their motion, the cells, therefore, predominantly propagate backward ($\Delta\Theta_{eff} > 90°$ in Fig. 2f). At low obstacle densities ($\rho = 2\%$ and $\rho = 4\%$), sporadic cell–obstacle interactions are sufficient to rectify the cells' motion chirality (Fig. 2b), thus effectively making them propagate forward ($\Delta\Theta_{eff} < 90°$ in Fig. 2f), consistently with the observed enhancement in $L_{eff}$ and $V_{eff}$ (Fig. 1c, d). While both $L_{eff}$ and $\Delta\Theta_{eff}$ point towards a minor rectification of the bacterial chirality for $\rho = 6\%$ and $\rho = 8\%$, $V_{eff}$ is comparable with the value

on the smooth surface as a consequence of an increased residence time due to cells stopping at the obstacles (Fig. 2c). For even higher densities (Fig. 2d, e), more frequent encounters with the obstacles increase the chances of cells turning backward and exiting near their entrance point, as also shown by $\Delta\Theta_{eff}$, once again, becoming comparable to the value on a smooth surface (Fig. 2f); $L_{eff}$ and $V_{eff}$ are, however, significantly reduced with respect to the values for $\rho = 0\%$ as cell–obstacle interactions physically hinder cell propagation on the surface in space and time.

**Forward scattering versus tumble-collisions.** When observing the trajectories in Fig. 2 and Supplementary Fig. 2, we can qualitatively identify two repeated types of cell–obstacle interactions, which we respectively named "forward scattering" and "tumble-collisions" (Fig. 3a, b). Quantitatively, these two classes of interactions can be distinguished based on an automated analysis that detects differences in how the cells' instantaneous speed $v$ and

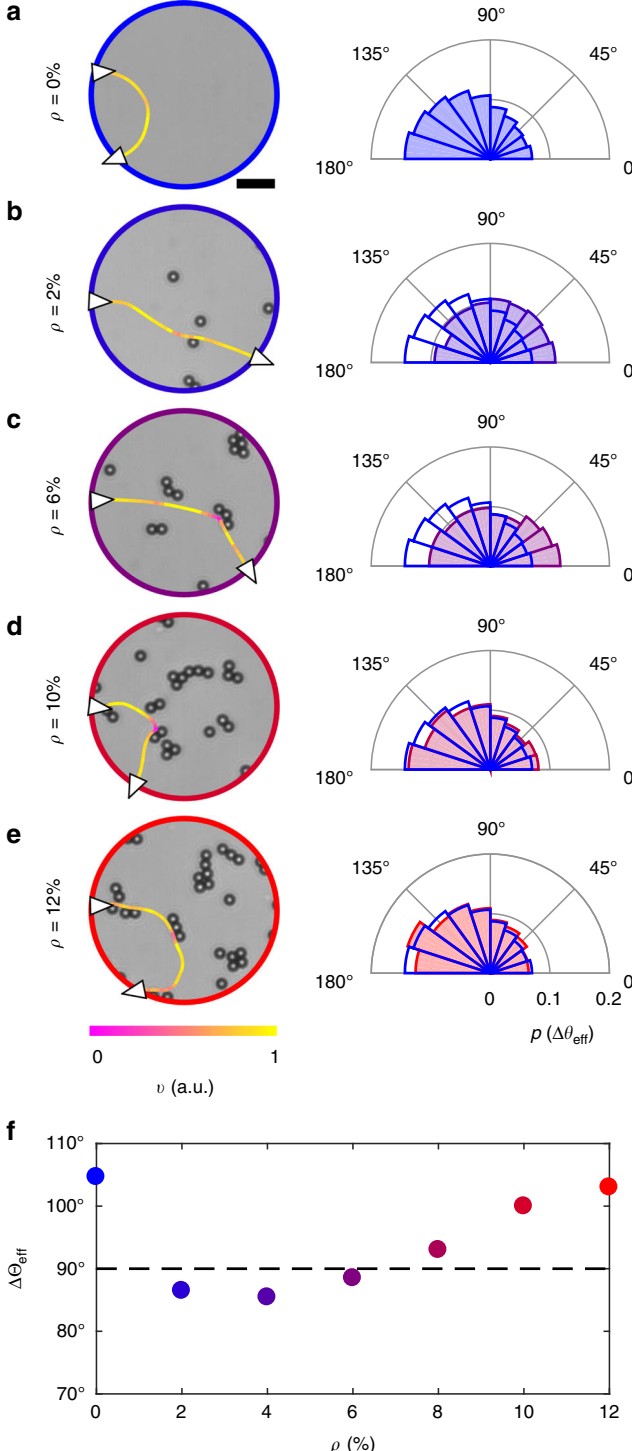

**Fig. 2** Change in effective propagation direction for *E. coli* cells near surfaces with micro-obstacles. **a–e** Exemplary trajectories and probability distributions of the change in effective propagation direction $\Delta\theta_{\text{eff}}$ for *E. coli* cells swimming through a circular area of radius $R = 25\,\mu\text{m}$ for different obstacle densities $\rho$: **a** $\rho = 0\%$, **b** $\rho = 2\%$, **c** $\rho = 6\%$, **d** $\rho = 10\%$ and **e** $\rho = 12\%$. The white triangles on the trajectories represent the direction of motion when entering and exiting the circular area, while the colour code of the trajectories represents the cells' instantaneous velocity $v$ normalised to its maximum value. The black scale bar in **a** corresponds to 10 μm. Each distribution is obtained from at least 1000 different trajectories, and $\Delta\theta_{\text{eff}} = 90°$ separates between forward ($\Delta\theta_{\text{eff}} < 90°$) and backward ($\Delta\theta_{\text{eff}} > 90°$) propagation. For reference, the distribution in **a** is also shown in **b–e** as a solid line. **f** Average change in effective propagation direction $\Delta\Theta_{\text{eff}} = \langle\Delta\theta_{\text{eff}}\rangle$ as a function of $\rho$ calculated from the previous probability distributions. The dashed line at 90° represents the separation between forward and backward propagation

spend a relatively long time at the obstacles before leaving, typically in a different (mainly backward) direction from that of arrival.

We can quantify these observations by calculating three quantities during a cell–obstacle interaction: the relative change in speed $\tilde{v} = \frac{v_{\text{int}}}{v_{\text{run}}}$ (Fig. 3c), where $v_{\text{int}}$ and $v_{\text{run}}$ are the average cell's speed during the interaction and the preceding run phase, the change $\Delta\theta_{\text{int}}$ in the cell's direction of motion pre- and post-interaction (Fig. 3d), and the interaction duration $t_{\text{int}}$ (Fig. 3e).

For tumble-collisions, $\tilde{v}$ is almost uniformly distributed in the range [0, 1] ($\langle\tilde{v}\rangle \approx 0.61$), $\Delta\theta_{\text{int}}$ shows a preference for cells leaving the obstacles in the opposite direction from that of approach, and $t_{\text{int}}$ follows a Poissonian distribution with a characteristic time ($\lambda_c \approx 1.33\,\text{s}$) comparable to the characteristic time of *E. coli* cells' tumbling[9]. In a tumble-collision, therefore, the bacteria tend to stop at the obstacle until a tumble event points them away from it, thus validating the decrease in $V_{\text{eff}}$ at high $\rho$ (Fig. 1) as jointly due to a decrease in the cells' propagation distance $L_{\text{eff}}$ and an increase in their residence time due to the presence of obstacles. This type of interaction becomes increasingly detrimental at higher obstacle densities as tumble-collisions become more probable (Supplementary Fig. 3f), also because of colloids forming larger clusters (Fig. 2 and Supplementary Fig. 2).

Contrarily, for forward scattering, $\tilde{v}$ follows a Gaussian distribution centred at $\langle\tilde{v}\rangle \approx 1$, $\Delta\theta_{\text{int}}$ is strongly peaked forward, and the cells quickly leave the obstacles as $t_{\text{int}}$ follows a negative exponential distribution with a characteristic time ($\lambda_{\text{fs}} = 0.29\,\text{s}$) comparable to the time needed for the average cell to travel a distance equal to one obstacle's diameter. In a forward-scattering event, therefore, the cells' speed and directionality are, on average, not significantly influenced by the obstacle during the interaction[45]. However, when leaving the obstacle, the cells' motion properties change: while the average translational speed ($v_{\text{fs}} = 12 \pm 4\,\mu\text{m s}^{-1}$) only mildly increases with respect to the value at $\rho = 0\%$, the cells' average angular speed is significantly reduced, i.e. on average, the cells' motion becomes significantly less chiral. Figure 3f shows the decorrelation of the cell's direction of motion $\theta$ over time calculated as

$$\langle\cos(\Delta\theta(\tau))\rangle = \langle\cos(|\theta(t_0 + \tau) - \theta(t_0)|)\rangle, \quad (1)$$

where $\langle\ldots\rangle$ represents an ensemble average and $t_0$ is the first instant following the end of a cell–obstacle interaction (Methods). By fitting Eq. (1) to the function $f(\tau) = \cos(\Omega\tau)e^{-\tau/\tau_0}$ (Methods), we can indeed appreciate how, after forward scattering, the cells' average angular speed $\langle\Omega\rangle$ is reduced to $\Omega_0 = 0.62\,\text{rad s}^{-1}$ from $\Omega_\infty = 0.81\,\text{rad s}^{-1}$ at $\rho = 0\%$ without, nevertheless, affecting the cell's motion persistence time ($\tau_0 \approx 3.5\,\text{s}$ in both cases). We thus hypothesise that forward scattering, through this chirality

direction of motion $\theta$ change near the obstacles (Supplementary Fig. 3a–d and Methods). Their detailed analysis offers a microscopic explanation for the previous experimental observations (Figs. 1 and 2). During forward-scattering events (Fig. 3a and Supplementary Fig. 3a, c), cells tend to approach the obstacles almost tangentially (Supplementary Fig. 3e) and their trajectories show minimal changes in speed and direction of motion, consistently with previous theoretical proposals[45]. Instead, during tumble-collisions (Fig. 3b and Supplementary Fig. 3b, d), more cells tend to approach the obstacles nearly head-on (Supplementary Fig. 3e), their speed drops significantly and they tend to

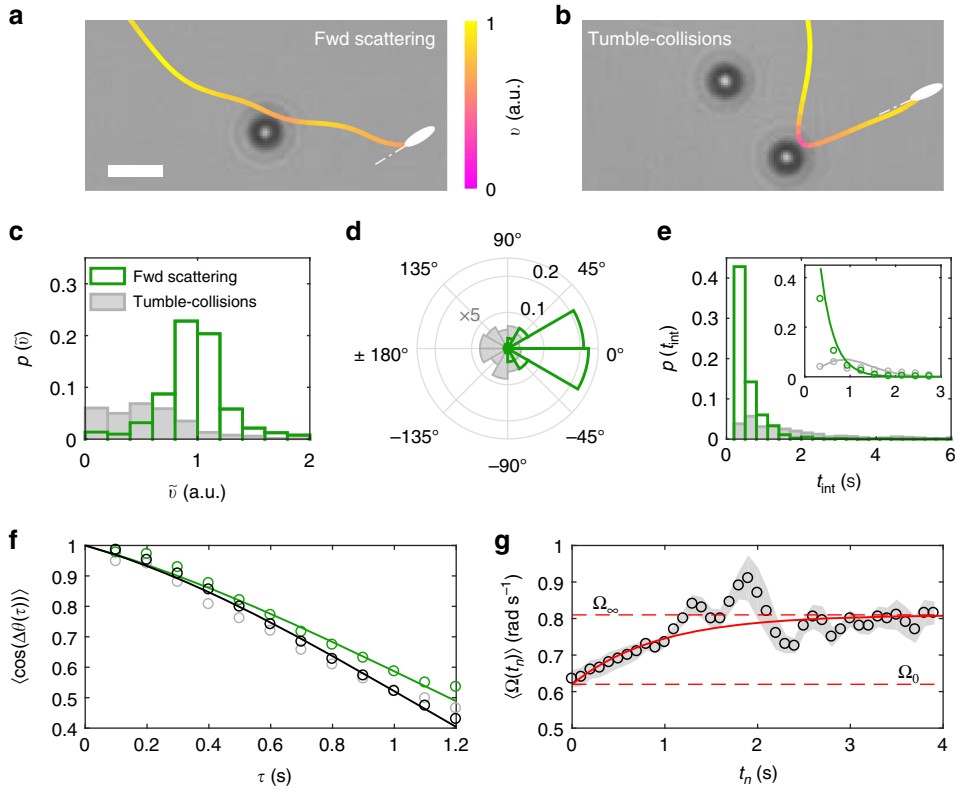

**Fig. 3** Differences between cell–obstacle interactions classified as forward scattering and tumble-collisions. **a, b** Examples of a forward-scattering event (**a**) and a tumble-collision (**b**). The stylised cells represent the trajectories' final position and orientation. The trajectories' colour code represents the cells' normalised instantaneous velocity $v$. The white scale bar corresponds to 5 μm. **c–e** Probability density distributions of the cells' **c** change in relative speed $\tilde{v}$, **d** change in direction of motion $\Delta\theta_{int}$, and **e** time $t_{int}$ spent at the obstacles during interaction for forward scattering (empty histograms) and tumble-collisions (filled histograms). All distributions are normalised to the total number of interactions to show the relative weight between forward-scattering events and tumble-collisions. In **d**, the distribution for tumble-collisions is 5× bigger for visualisation. In the inset in **e**, the experimental distributions (circles) are fitted to an exponential distribution for forward scattering and to a Poissonian distribution for tumble-collisions (solid lines). **f** Average experimental decorrelation $\langle\cos(\Delta\theta(\tau))\rangle$ of the cells' direction of motion over time for forward scattering (green circles) and tumble-collisions (grey circles) calculated as ensemble average from the first instant $t_0$ after at least 100 cell–obstacle interactions. For reference, the black circles show the same quantity calculated in the absence of obstacles (Eq. (3), Methods). The solid lines are fittings to the function $f(\tau) = \cos(\Omega\tau)e^{-\tau/\tau_0}$. **g** Time evolution of the cells' average angular speed $\langle\Omega\rangle$ (black circles) after the end of a forward-scattering event. Each value is calculated as an ensemble average from the $n$-th instant $t_n$ after at least 100 forward-scattering events. The shaded area represents one standard deviation around the average values. The solid line represents the fitting to the function $f(t_n) = \Omega_\infty - (\Omega_\infty - \Omega_0)e^{-t_n/\tau_\Omega}$, where $\Omega_0$ and $\Omega_\infty$ (dashed lines) are the average angular speeds at $t_0$ and for $\rho = 0\%$, respectively

rectification, is the microscopic reason behind the increase in $L_{eff}$ and $V_{eff}$ observed in Fig. 1 at small $\rho$, when this type of interaction is indeed predominant (Supplementary Fig. 3f). Practically, this rectification is due to an average increase of the cells' distance from the closest surface because of a hydrodynamic torque experienced when swimming near the obstacles (Supplementary Fig. 4a, b)[14]. It is important to note that this is an average behaviour as, depending on which side the cells pass the obstacle, not all forward-scattering events will lead to a change in height (Supplementary Fig. 4c). Interestingly, after tumble-collisions, the cells behave similarly to those swimming without obstacles (Fig. 3f), thus further confirming that, during tumble-collisions, the bacteria tend to stop at the obstacles before restarting their motion on the surface. Figure 3g shows how $\langle\Omega\rangle$ changes as the cells move away from the obstacles, gradually restabilising at $\Omega_\infty$ from $\Omega_0$ following the exponential trend

$$\langle\Omega(t_n)\rangle = \Omega_\infty - (\Omega_\infty - \Omega_0)e^{-t_n/\tau_\Omega}, \qquad (2)$$

where $t_n$ is the $n$-th instant following the end of a forward-scattering event and $\tau_\Omega = 0.93$ s (as fitted from the experimental data). In fact, as the cell changes its height, it approaches the sample chamber's other surface where it gets entrapped again

(after a wobbling period[30]) until another forward-scattering event, or an out-of-plane tumble, induce a new change in height (Supplementary Fig. 4). In our experimental configuration, therefore, the effect of a forward-scattering event on the cell's motion is over after the cell has moved away from the obstacle by a distance $\ell_{int} = v_{fs}\tau_\Omega \approx 11\,\mu m$, on average. Forward scattering also influences the cells' motion near the surface in thicker sample chambers (Fig. 4). In this case, individual forward-scattering events on the obstacles lead to an increased probability for the cells to detach from the surface with respect to the case for $\rho = 0\%$ (Fig. 4a) as also shown by the examplary trajectories in Fig. 4b, c. This probability almost doubles with respect to the homogenous case in the density range between $\rho = 2\%$ and $\rho = 8\%$ due to forward scattering (Fig. 4a, c) and, only for $\rho > 8\%$, the chances of detachment reduce with respect to the lower density values due to tumble-collisions (Fig. 4a, d).

**Mechanism underlying the cells' enhancement in propagation.** To test the relative importance of forward-scattering events versus tumble-collisions in determining the non-monotonic trends of $L_{eff}$ and $V_{eff}$ with increasing $\rho$, we considered a simple particle-

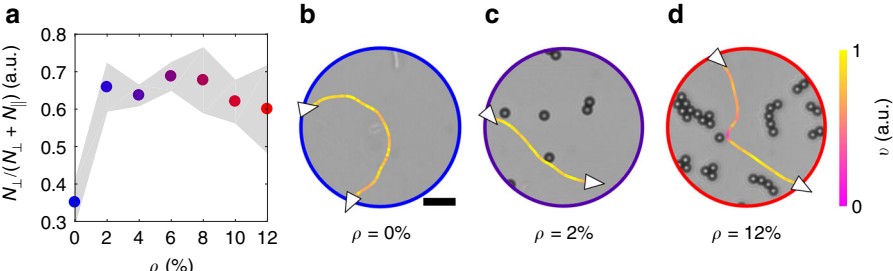

**Fig. 4** Probability of cell detachment from the surface as a function of obstacle density. **a** Probability of detachment from the surface as a function of the obstacle density $\rho$ for *E. coli* cells swimming through a circular area of radius $R = 25\,\mu m$. This probability is calculated by considering all cells' trajectories that enter the circular area through its perimeter (i.e. that are entrapped at the surface) and leave it either through its perimeter (i.e. still entrapped at the surface, $N_{\parallel}$) or by moving out of plane ($N_{\perp}$). The samples are analogous to those in Fig. 1 (Methods) with sparse 10-μm polystyrene particles as spacers. Each value is obtained by averaging over at least five independent experiments. The shaded area around the average values represents one standard deviation. In each independent experiment, at least 70 different trajectories were used to determine the probability of detachment from the surface for every value of $\rho$. **b–d** Exemplary trajectories showing *E. coli* cells that **b** remain entrapped at the surface in the absence of obstacles, **c** detach from the surface after a forward-scattering event, and **d** remain entrapped at the surface after a tumble-collision. The white triangles on the trajectories represent the direction of motion when entering and leaving the circular area either through the perimeter (i.e. still entrapped at the surface) or by moving out of plane, while the colour code of the trajectories represents the cells' instantaneous velocity $v$ normalised to its maximum value. The black scale bar in **b** corresponds to 10 μm

based model that includes the two types of cell–obstacle interactions (Methods). Briefly, cells are modelled as chiral active particles, where the angular speed $\Omega$ depends on the distance to the closest obstacle (forward scattering) and the direction of motion is changed at random when the particle's speed drops significantly (tumble-collision). Initially, we consider the individual obstacles distributed at random without overlap (Supplementary Fig. 5a). Figure 5a shows a good agreement between the experimental and simulated values of $L_{\text{eff}}$, $V_{\text{eff}}$ and $\Delta\Theta_{\text{eff}}$. In particular, the simulated distributions of the change in effective propagation direction $\Delta\theta_{\text{eff}}$ (Supplementary Fig. 6a) confirm that the enhancement in $V_{\text{eff}}$ at low obstacle densities is due to the rectification of the active particles' chirality by the interaction with the obstacles. Interestingly, the experimental behaviour in Figs. 1 and 2 is qualitatively preserved even when only considering forward-scattering events and excluding tumble-collisions (Fig. 5b and Supplementary Fig. 6b): a few micro-obstacles enhance the particles' propagation with respect to a smooth surface before hindering it at higher densities; however, without the further penalisation introduced by tumble-collisions, significant localisation effects only appear at slightly higher obstacle densities than they would when tumble-collisions are considered. These numerical results, therefore, show how forward scattering is the primary mechanism of particle–obstacle interaction behind the non-monotonic trends of $L_{\text{eff}}$ and $V_{\text{eff}}$ with increasing $\rho$, with tumble-collisions mainly influencing this behaviour quantitatively rather than qualitatively. Without this mechanism, $L_{\text{eff}}$ and $V_{\text{eff}}$ decrease monotonically with the density of obstacles as the particles get increasingly reflected backward by their presence due to the repulsion term (Fig. 5c and Supplementary Fig. 6c), with tumble-collisions playing again a primarily qualitative role (Supplementary Fig. 7).

To test the robustness of our experimental results with respect to how the obstacles are distributed on the surface, we also simulated the motion of chiral active particles moving through obstacles arranged according to a triangular lattice (Supplementary Fig. 5b and Methods) and through a random distribution of non-overlapping trimers (Supplementary Fig. 5c and Methods). In these simulations, the interactions with the obstacles include all three cell–obstacle interaction terms (Methods). Overall, our simulations show that the enhancement in the propagation of chiral active particles near a surface by an optimal low density of obstacles is a robust observation, which is qualitatively

independent from the obstacle distribution (Figs. 5a and 6). For obstacles consisting of individual particles (Supplementary Fig. 5a, b and Methods), forward propagation is enhanced over a larger range of obstacle densities when obstacles are distributed according to a periodic lattice (Fig. 6a) rather than an uncorrelated distribution (Fig. 5a). Due to the periodicity of the lattice, obstacles cannot be clustered together at low densities and the likelihood of observing tumble-collisions is lower with most particle–obstacle interactions leading to forward-scattering events (Supplementary Figs. 6a and 8a). Tumble-collisions instead tend to be favoured by random configurations of obstacles due to localisation phenomena. The size of the clusters is also an important parameter. For a given density $\rho$ of randomly distributed clusters (Supplementary Figs. 5a, c), forward propagation is enhanced by bigger clusters (Fig. 6b) rather than by smaller clusters (Fig. 5a). The chances of being reflected back are indeed lower with bigger clusters (Supplementary Figs. 6a and 8b) as these occupy the available space less evenly than isolated obstacles, thus decreasing the odds for a cell to interact with an obstacle during a run.

**Scaling behaviour over swimming distance.** Finally, Fig. 7 shows how the behaviour observed in Figs. 1 and 2 is preserved over large propagation distances, both in experiments and simulations. The enhancement of the average effective propagation speed $V_{\text{eff}}$ at low obstacle densities can be observed across all areas whose diameter is larger than the average radius of curvature $R_{\text{EC}}$ of the chiral bacterial cells (Fig. 7a, b). For very small areas indeed ($R = 5\,\mu m$, i.e. $2R < R_{\text{EC}}$), cells propagate better in the absence of obstacles since these, like for non-chiral active colloids[41], disrupt their motion which is mainly directed forward (Fig. 7c and Supplementary Fig. 9a). However, when $R = 10\,\mu m$ ($2R > R_{\text{EC}}$), the values of $V_{\text{eff}}$ at $\rho = 0\%$ and $\rho = 2\%$ become comparable (Fig. 7a). For increasing $R$ values, a clear peak in $V_{\text{eff}}$ can be observed around $\rho = 2\%$ (Fig. 7a, b) due to the rectification of the cells' chirality by the obstacles as shown by the persistent minimum in $\Delta\Theta_{\text{eff}}$ (Fig. 7c): even for $R = 50\,\mu m$ (i.e. when the area is approximately two orders of magnitude bigger than the typical cell's size), $V_{\text{eff}}$ at $\rho = 2\%$ is ≈20% higher than at $\rho = 0\%$ and the distribution of $\Delta\theta_{\text{eff}}$ is more uniform than at any other $\rho$ value where these distributions are peaked backward (Supplementary Fig. 9b). This long-range enhancement in cells' propagation due

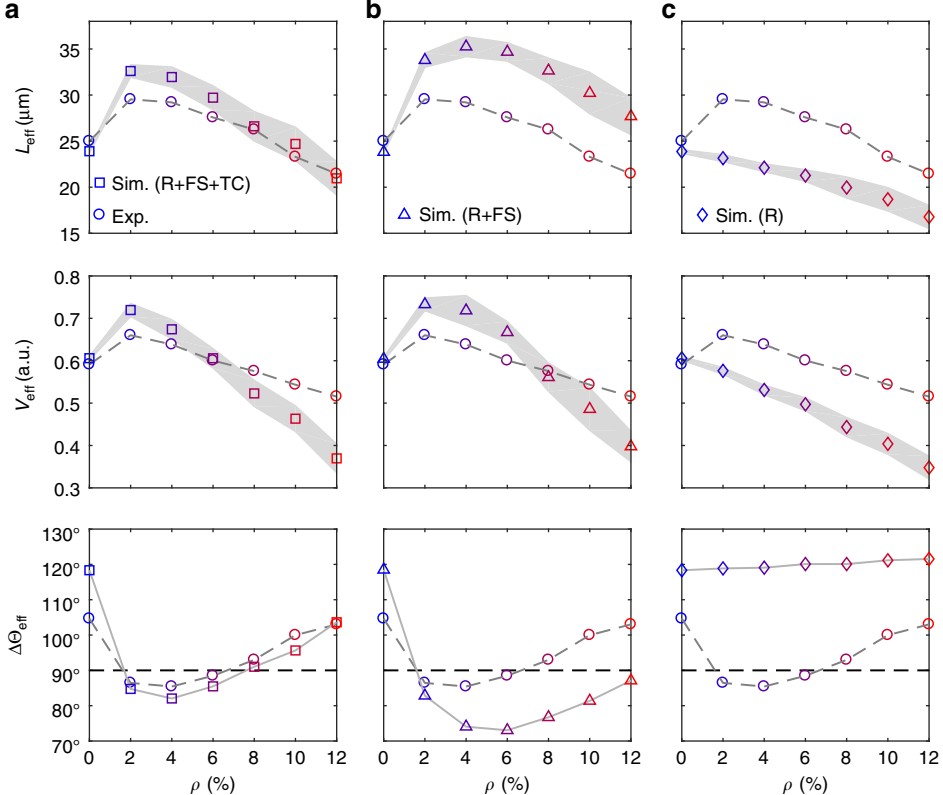

**Fig. 5** Comparison between experiments and numerical simulations. **a–c** Simulated average effective propagation distance $L_{eff}$, normalised average effective propagation speed $V_{eff}$ and average change in effective propagation direction $\Delta\Theta_{eff}$ as a function of the obstacle density $\rho$ for chiral active particles self-propelling through a circular area of radius $R = 25\,\mu m$ containing obstacles distributed at random without overlap (Supplementary Fig. 5a). The particles self-propel in the presence of **a** all three cell–obstacle interaction terms (R: repulsive interaction; FS: forward scattering; TC: tumble-collisions), **b** without tumble-collisions (TC) and **c** with repulsion (R) alone (Methods). Each value is obtained from averaging over 3000 different trajectories. The shaded area around the mean values of $L_{eff}$ and $V_{eff}$ represents one standard deviation. The solid line connecting the values of $\Delta\Theta_{eff}$ is a guide for the eyes. The corresponding probability distributions of the change in effective propagation direction $\Delta\theta_{eff}$ are shown in Supplementary Fig. 6. The corresponding experimental values (Figs. 1 and 2) are shown for reference (circles). Supplementary Figure 7 shows simulations where only repulsion (R) and tumble-collisions (TC) are considered

to a few obstacles is also confirmed by the higher value of the measured translational diffusion coefficient $D$, as estimated from the asymptotic behaviour of the cells' mean square displacement (Fig. 8 and Methods): when compared to a smooth surface, the cell's diffusivity is indeed enhanced by a factor $\frac{D_{2\%}}{D_{0\%}} = 1.55$ ($D_{0\%} = 42.82\,\mu m^2\,s^{-1}$ and $D_{2\%} = 66.58\,\mu m^2\,s^{-1}$).

## Discussion

Our results demonstrate the critical role played by surface defects on the near-surface swimming of bacterial cells. In particular, we show how cells' propagation near surfaces is significantly enhanced by individual forward-scattering events due to a few microscopic obstacles of size comparable to the typical bacterial cell. The intuitive behaviour, where obstacles hinder propagation rather than enhancing it[31,40,41], is only recovered at higher obstacle densities due to cells' head-on tumble-collisions with the obstacles.

As the enhancement in cells' propagation at low obstacle densities is hydrodynamic in nature, obstacle size is of paramount importance. On the one hand, much bigger obstacles (i.e. approximately one order of magnitude bigger than the typical bacterial cells' size) can lead to cells being hydrodynamically trapped in circular trajectories around the obstacles for long times[25,26,45]. On the other hand, smaller obstacles than those used here will produce less hydrodynamic torque on the

swimming bacteria, thus diminishing the strength of forward-scattering events. In realistic situations, obstacles can be expected to vary in size, shape and density so that all the previous mentioned effects (e.g. forward-scattering, tumble-collisions, entrapment) can in principle influence cells' propagation on surfaces simultaneously.

Our results are corroborated by a numerical model based on chiral active Brownian particles cruising through micro-obstacles that confirms the universality of the experimentally observed behaviour. This model highlights how the interaction with a few obstacles enhances particles' propagation on surfaces as long as two main factors are present: chirality in the particles' motion and a partial correction of such chirality during the repulsive inter-action with the obstacles. Overall, our numerical results suggest that the experimentally observed behaviour should be independent, at least qualitatively, of the microscopic nature of the self-propulsion mechanism and of the repulsive interaction between particles and obstacles as long as the two previous conditions are satisfied. Undoubtedly, further surface motility experiments are required to understand to what extent these two conditions apply to bacterial swimming mechanisms other than the run-and-tumble of peritrichously flagellated *E. coli* cells as well as to test how the qualitative and quantitative nature of the cell–obstacle interaction changes with the swimming mechanism and the mechanism used by the cells to change direction of motion[46]. When tumble-collisions are included, our simplified model with

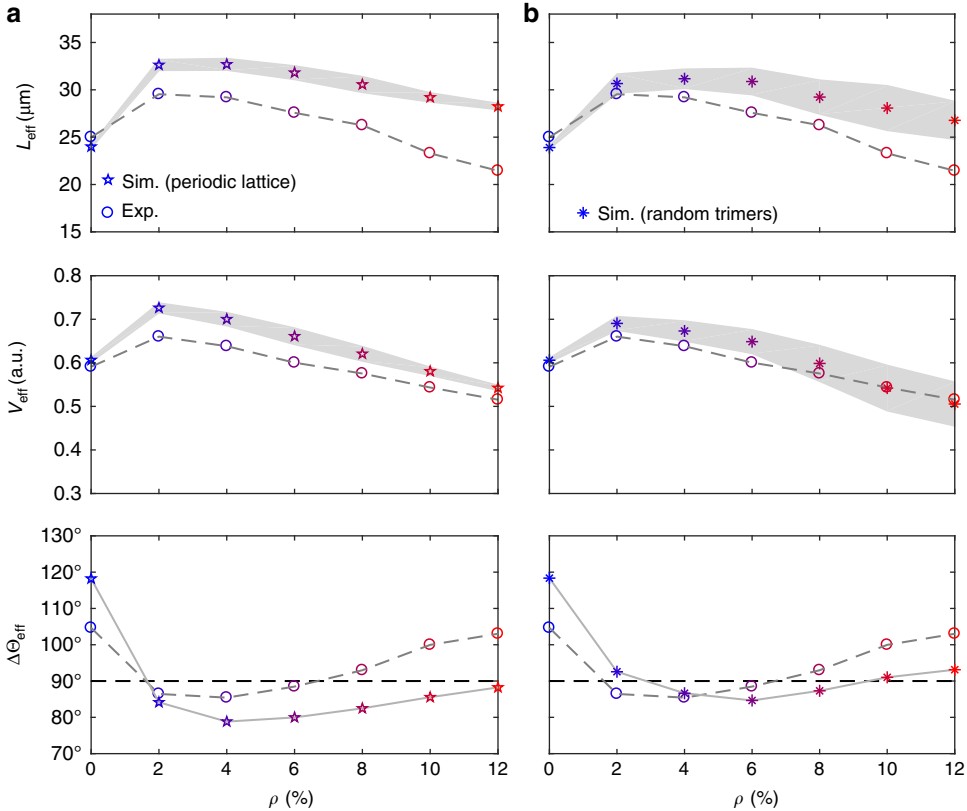

**Fig. 6** Influence of obstacle distribution. **a**, **b** Simulated average effective propagation distance $L_{eff}$, normalised average effective propagation speed $V_{eff}$ and average change in effective propagation direction $\Delta\Theta_{eff}$ as a function of the obstacle density $\rho$ for chiral active particles self-propelling through a circular area of radius $R = 25\,\mu m$ containing obstacles distributed according to a triangular periodic lattice (**a**) (Supplementary Fig. 5b and Methods) and a random distribution of non-overlapping trimers (**b**) (Supplementary Fig. 5c and Methods). The interactions with the obstacles include all three cell–obstacle interaction terms: repulsive interactions, forward-scattering events and tumble-collisions (Methods). Each value is obtained from averaging over 3000 different trajectories. The shaded area around the average values of $L_{eff}$ and $V_{eff}$ represents one standard deviation. The solid line connecting the values of $\Delta\Theta_{eff}$ is a guide for the eyes. The corresponding probability distributions of the change in effective propagation direction $\Delta\theta_{eff}$ are shown in Supplementary Fig. 8. The corresponding experimental values (Figs. 1 and 2) are shown for reference (circles)

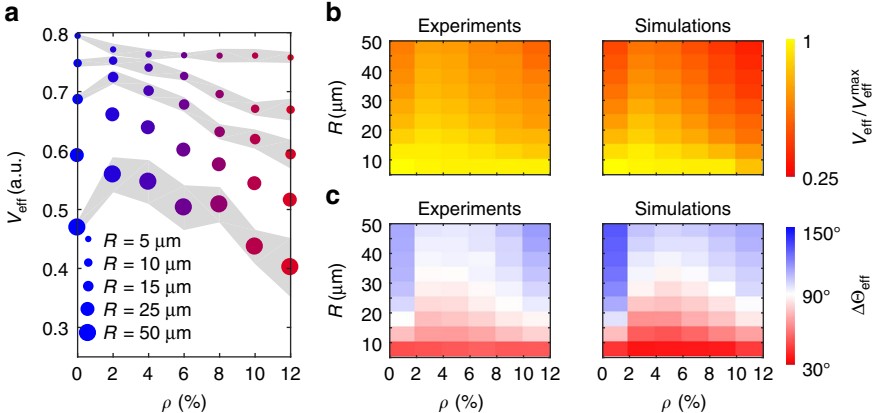

**Fig. 7** Scaling behaviour of chirality rectification in space. **a** Experimental average effective propagation speed $V_{eff}$ as a function of the obstacle density $\rho$ for circular areas of increasing radius $R$. Each value is obtained from averaging over at least 200 different trajectories. The shaded areas around the average values represent one standard deviation. The case for $R = 25\,\mu m$ (Fig. 1d) is also shown for reference. **b** Average effective propagation speed $V_{eff}$ and **c** average change in effective propagation direction $\Delta\Theta_{eff}$ as a function of $\rho$ and $R$ in experiments and simulations. $V_{eff}$ is normalised to its maximum values $V_{eff}^{max}$ for visualisation purposes ($V_{eff}^{max} = 0.79$ in experiments and $V_{eff}^{max} = 0.95$ in simulations)

spherical particles can reproduce the main experimental observations obtained with *E. coli* cells in a close-to-quantitative fashion. In principle, the quantitative match between our experimental observations and numerical results can be improved

further by taking into account the actual cell's shape and exact swimming mechanism.

Soft-lithography techniques can also be employed to fabricate obstacles on surfaces with improved control over their size and

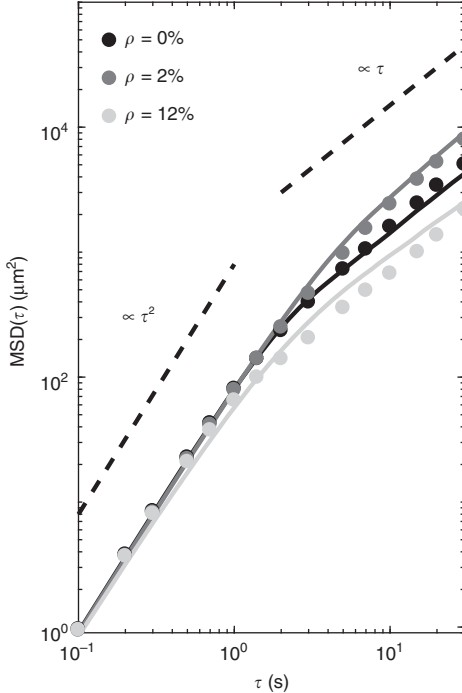

**Fig. 8** *E. coli* cells' mean square displacement for different obstacle densities. Average mean square displacements (MSDs) of *E. coli* cells swimming on a smooth surface ($\rho = 0\%$), in the presence of a few obstacles ($\rho = 2\%$) and at higher obstacle densities ($\rho = 12\%$) in experiments (circles) and simulations (solid lines). The MSD at $\rho = 2\%$ shows a clear enhancement in diffusivity for the cells when compared to the MSD at $\rho = 0\%$. The MSDs calculated from simulated trajectories match well the experimental ones. Both in experiments and simulations, the MSD at $\rho = 12\%$ shows a decrease in diffusivity when compared to the MSD at $\rho = 0\%$. This decrease is lower in simulations rather than in experiments as our model does not account for the fact that, in experiments, cells can stop at an obstacle for a prolonged period of time, thus inducing a stronger transient subdiffusive behaviour. The two dashed lines, respectively, show ballistic ($\propto \tau^2$) and diffusive ($\propto \tau$) behaviour for reference. Each experimental MSD curve was obtained as an ensemble average over at least 30 trajectories (each at least 30 s long), while each simulated MSD curve was calculated as an ensemble average over 20,000 trajectories (each 30 s long) obtained from 200 different obstacle configurations with 100 non-interacting particles each (Methods)

distribution, thus enabling a quantitative study of how these parameters influence the position and the width of the experimentally observed peak in effective velocity with obstacle density. For example, in the presence of high densities of periodic obstacles ($\rho > 12\%$), forward-scattering events could amplify cell propagation if the spacing between the obstacles became comparable to the cells' characteristic run length due to cells being channelled by the periodic lattice[38,39,47].

Interestingly, for *E. coli* cells, as a consequence of a hydrodynamic torque, forward-scattering events on the obstacles also lead the cells' trajectory to leave the surface. Along with the intermittent motion shown by some pathogenic strains of *E. coli* near a flat surface[48], this behaviour can thus offer a way to potentially reduce escape times when swimming near it and maximise near-surface diffusivity[23–26]. As our study focused on flat surfaces, promising future directions include testing the robustness of the identified forward-scattering mechanism on curved surfaces (where the surface curvature varies on a length scale comparable to the cells' persistence length), near interfaces in the presence of floating obstacles as well as in 3D porous structures.

We envisage our results will help understand the individual and collective behaviour of chiral active matter in complex and crowded environments at all length scales[16]: examples include other microorganisms, such as microalgae and sperm cells[49,50], and macroscopic robotic swarms[51]. Another problem of fundamental interest is to understand how both motion chirality and long interaction times at high obstacle densities influence the invariance of the effective residence time within a region predicted for purely diffusive random walkers[44] and recently verified for non-chiral bacteria[52]. Beyond these fundamental interests, our finding can help design microfluidic devices to sort and rectify chiral active matter[16,18,53–55]. Similarly, microstructured surfaces can be employed to better understand the emergence of bacterial social behaviours in natural habitats and to devise engineered materials to control and prevent bacterial adhesion to surfaces.

## Methods

**Bacterial culture and preparation**. Motile *E. coli* cells (wild-type strain RP437, *E. coli* Genetic Stock Center, Yale University) were first revived from a $-80\,°C$ stock by incubating at $37\,°C$ overnight on Tryptic Soy agar (TSA, Sigma-Aldrich). Using aseptic technique, a single colony was then picked and grown for 18 h at $37\,°C$ in 50 mL Tryptone Soy broth (TSB, Sigma-Aldrich) in a conical flask shaking at 150 rpm. The culture was then diluted 1:100 into fresh TSB and incubated again for 4 h 20 min at $37\,°C$ while shaking at 150 rpm until the culture reached its mid-log phase at a point where the bacteria were experimentally found to be most motile ($OD_{600}$ ~1.4). Subsequently, 0.1 ml of this dilution was centrifuged at 750 rpm at room temperature for 5 min. Finally, the supernatant was removed and the resulting precipitated bacterial cell pellets were gently resuspended in 0.1 ml of motility buffer containing 10 mM monobasic potassium phosphate ($KH_2PO_4$, Sigma-Aldrich), 0.1 mM EDTA (pH 7.0, Promega), 10 mM dextrose ($C_6H_{12}O_6$, Sigma-Aldrich) and 0.002% of Tween 20 (Sigma-Aldrich). This process was repeated three times in order to completely replace the growth medium with motility buffer and halt bacterial growth. The final bacterial suspension was either used directly for the high-concentration experiments in Fig. 1e or diluted 1:10 elsewhere. The first time we prepared the sample from the purchased strain, we introduced an additional step to select the most motile bacteria by inoculating 5 µL of the 1:100 dilution in the centre of a soft TSA plate (0.3% agar)[56]; this plate was then incubated at $37\,°C$ overnight. The following day, 5 µL of soft agar and bacteria were picked from the edge of the colony formed on the plate and inoculated at the centre of a new soft agar plate. After repeating this procedure three times, a stock solution of the third generation of bacteria was prepared in 50 mL of TSB with the addition of 10%(v/v) glycerol (Sigma-Aldrich) and stored at $-80\,°C$. This stock solution was used as the starting point for all experiments.

**Sample preparation**. Each experiment was performed in a homemade sample chamber formed by a clean microscope glass coverslip as the upper boundary and a clean microscope slide as the lower boundary. The coverslip and the slide were cleaned by sequentially sonicating them in acetone (>99.8%), ethanol (>99.8%) and deionised (DI) water (resistivity >18 MΩ cm) for 5 min each. After cleaning, 5 µL of a 0.25 wt% water suspension of polystyrene microparticles (diameter $d = 2.99 \pm 0.07$ µm, microParticles GmbH) containing 0.1 M sodium chloride (NaCl) was left to evaporate on the clean slide, thus depositing clusters of particles on the glass surface. By placing the slide on a hotplate heated to $160\,°C$ (well below the polystyrene melting temperature of ≈$240\,°C$) for 5 min, we improved the long-term adhesion of these clusters to the glass surface without deforming the particles because of melting. Remaining salt crystals and colloids that did not strongly adhere were washed away with DI water before drying the slide with nitrogen gas. Following this protocol, we were able to produce random distributions of fixed obstacles with different density values, $0\% \le \rho \le 12\%$, on the same surface, where $\rho$ is the fractional surface coverage of the colloids in a given region of interest (typically circular with radius $R$ in our experiments). Finally, 10 µL of the bacterial suspension was deposited on the glass slide, which was subsequently sealed with the clean coverslip to form a chamber with spacing provided by the same colloidal particles. The size of the polystyrene microparticles was indeed chosen to guarantee, after sealing the chamber, a quasi-2D geometry for the bacteria to move in without the possibility of squeezing through the remaining gaps between two colloids in contact.

**Experimental setup**. All experimental observations were performed on a homemade inverted bright-field microscope enclosed in a custom-made environmental chamber (Okolab) with temperature control ($T = 22 \pm 0.5\,°C$). The microscope was mounted on a floated optical table for vibration dampening. The bacteria were tracked by digital video microscopy using the image projected by a microscope objective (×20, NA = 0.5, Nikon CFI Plan Fluor) on a monochrome CMOS camera (1280 × 1024 pixels, Thorlabs DCC1545M) at 10 f.p.s.[57]. The magnification of our imaging path allowed us to achieve a conversion of 0.22 µm per pixel,

corresponding to a field of view of $\approx 280 \times 225\ \mu m^2$. The incoherent illumination for the tracking of the bacteria was provided by a red LED ($\lambda = 660$ nm, Thorlabs M660L3-C2) employed in a Köhler configuration to control and improve coherence and contrast of the illumination at the sample plane. The typical duration of an experiment was $\approx 60$ min before bacteria motility started to decrease considerably. In total, we recorded over 3500 individual bacterial trajectories of variable duration. The data shown in the figures are obtained from the analysis of segments of these trajectories.

**Estimation of the cells' average speeds**. We estimated the average translational speed, $\langle v \rangle$, and the average angular speed, $\langle \Omega \rangle$, of the bacterial cells by taking an average of the individual speeds of 85 trajectories obtained on a smooth surface, i.e. for $\rho = 0\%$ (Supplementary Fig. 1). To determine $\langle v \rangle$, we first calculated the probability distribution of the instantaneous speed $v$ for each trajectory, as exemplified in Supplementary Fig. 1a. This distribution typically shows two peaks which we were respectively able to predominantly assign to a cell's tumble phase and its run phase, so that, by thresholding at the local minimum between the two peaks, the average translational speed of each trajectory could be estimated from the speed values associated with the run phase. To do so, we first segmented each trajectory in runs separated by tumbles (inset in Supplementary Fig. 1a) following the procedure detailed in ref. [58]. Briefly, after smoothing each trajectory with a running average over 5 time steps, the duration of individual tumbles was determined based on two dimensionless thresholds ($\alpha = 0.7$ and $\beta = 2$), which were respectively used to determine sufficiently large local variations in instantaneous speed $v$ and direction of motion $\theta$. The numerical values of these two thresholds were validated against several trajectories by visual inspection. Similarly, to estimate $\langle \Omega \rangle$, we first calculated an angular speed $\Omega$ for each trajectory independently (Supplementary Fig. 1b) and then averaged these values over all 85 trajectories. In analogy to the estimation of the persistence length of a polymer[59], each $\Omega$ was determined from the decorrelation of the cell's direction of motion $\theta$ over time fitting the following expression to the function $f(\tau) = \cos(\Omega\tau)e^{-\tau/\tau_0}$,

$$\overline{\cos(\Delta\theta(\tau))} = \overline{\cos(|\theta(t+\tau) - \theta(t)|)}, \tag{3}$$

where $\Delta\theta$ is the angle between the tangents to the trajectory at times $t + \tau$ and $t$, the bar represents a time average, and $\tau_0$ is the trajectory's persistence time. The direction of motion therefore decorrelates following an exponential decay, which is modulated by a cosine function when $\Omega \neq 0$. Supplementary Fig. 1b shows exemplary fits to the experimental data for three different values of $\Omega$.

**Estimation of the cells' effective propagation quantities**. To calculate the average effective propagation quantities ($L_{eff}$, $V_{eff}$ and $\Delta\Theta_{eff}$) of the bacterial cells, we first divided the entire field of view of all acquired experimental videos into $M$ circular areas of radius $R$ with centres on a square lattice of periodicity $R$. For example, for $R = 25\ \mu m$ as in Fig. 1b, $M = 80$ in our field of view. For statistics, based on its calculated obstacle density value, each circular area was then mapped on a discrete $\rho$ scale with a $2 \pm 0.6\%$ separation step, and the trajectories contained within were used to calculate the average effective propagation quantities of the corresponding $\rho$ value on this scale. We excluded from the analysis all the trajectories ($\leq 5\%$ at any $\rho$) that did not exit a circular area after entering it and, to avoid biasing our results with extremely short trajectories, those that predominantly moved along the area perimeter, i.e. those that penetrated $\leq 10\%$ of the area diameter without interacting with any obstacle. After smoothing with a running average over 5 time steps, we assigned an effective propagation distance $\ell_{eff} = \|\mathbf{P}_{out} - \mathbf{P}_{in}\|$ to each of the remaining trajectories, where $\mathbf{P}_{in}$ and $\mathbf{P}_{out}$ are the trajectory's entrance and exit points, respectively (Fig. 1b). This distance can take any value between 0 (the cell exits from where it entered) and $2R$ (the cell exits at the diametrically opposite point from where it entered). By averaging $\ell_{eff}$ over all trajectories propagating through all circular areas of same $\rho$, we calculated the average effective propagation distance at different obstacle densities as $L_{eff} = \langle \ell_{eff} \rangle$. The normalised average effective propagation speed $V_{eff}$ as a function of $\rho$ was instead calculated as $V_{eff} = \langle \frac{\ell_{eff}}{v_{eff} t_{eff}} \rangle$, where, for a single cell, $v_{eff}$ and $t_{eff}$ are, respectively, its average translational speed when in run phase and its time of residence within the circular area. The normalisation by $v_{eff}$ makes different trajectories directly comparable, thus accounting for the fact that the intercell variability in translational speed can influence residence times. Finally, the average change in effective propagation direction as a function of $\rho$ was calculated as $\Delta\Theta_{eff} = \langle \Delta\theta_{eff} \rangle = \langle |\theta(t_{out}) - \theta(t_{in})| \rangle$, where $\Delta\theta_{eff}$ is the angle between the tangents to a cell's trajectory when exiting and entering a circular area, respectively.

**Classification of cell–obstacle interactions**. In order to distinguish between forward scattering and tumble-collisions, we first identified all cell–obstacle interactions along each trajectory. To simplify our analysis, we considered an interaction to take place only while there was a degree of overlap between the area occupied by an obstacle and the area occupied by the average cell body (centred along the trajectory and aligned with its direction of motion). Tumble-collisions were then identified out of this pool of interactions in analogy to the procedure for determining tumbles on a cell's trajectory as in Supplementary Fig. 1a[58]. Briefly, after smoothing each trajectory with a running average over 5 time steps, individual tumble-collision events were selected based on two concomitant dimensionless

thresholds ($\alpha = 0.7$ and $\beta = 2$), which were respectively used to determine sufficiently large local variations in instantaneous speed $v$ and direction of motion $\theta$ during the cell interaction with the obstacle with respect to the values preceding it (Supplementary Fig. 3). A first criterion set a threshold on the variation of instantaneous speed by detecting a local minimum in $v$ during the cell–obstacle interaction at a time $t_{min}$ (Supplementary Fig. 3a, b); the times $t_1$ and $t_2$ of the two closest local maxima in $v$ (Supplementary Fig. 3a, b) were then identified and used to compute the relative change in speed $\frac{\Delta v}{v(t_{min})}$, where $\Delta v = \max[v(t_1) - v(t_{min}), v(t_2) - v(t_{min})]$. A second criterion set a threshold on the variation of the direction of motion by first detecting a local maximum in the absolute value of the time derivative of $\theta$ during the cell–obstacle interaction at time $t_{max}$ (Supplementary Fig. 3c, d); the times $t_1$ and $t_2$ of the two closest local minima (Supplementary Fig. 3c, d) were then identified, and used to compute the cumulative change in direction during the interaction as $|\Delta\theta| = \sum_{t=t_1}^{t_2-1} |\theta(t+1) - \theta(t)|$. If both $\frac{\Delta v}{v(t_{min})} \geq \alpha$ and $|\Delta\theta| \geq \beta\sqrt{2D_{rot}(t_2 - t_1)}$ (with $D_{rot} = 0.1$ rad$^2$ s$^{-1}$ (ref. [58])) were satisfied, the cell–obstacle interactions were classified as tumble-collisions. All remaining interactions were classified as forward-scattering events. We determined that, following this protocol, $\approx 11\%$ of all interactions were wrongly attributed based on the visual inspection of 225 cell–obstacle interactions selected at random.

**Numerical model**. We consider a numerical model where identical spherical active particles of radius $d/2$ move inside a two-dimensional square box of side $B = R + 4.5\ \mu m$ with periodic boundary conditions, where $R$ is the variable radius of a circular area in the box centre. Within the circular area, we placed circular obstacles with variable densities $\rho$ deposited sequentially at random without overlap (Supplementary Fig. 5a), according to a periodic triangular lattice (lattice constant equal to 2.75$d$) where $\rho = 12\%$ corresponds to a complete lattice and lower obstacle densities are obtained by removing particles uniformly at random (Supplementary Fig. 5b), or sequentially as non-overlapping trimers (i.e. triangular clusters of obstacles) with a random orientation (Supplementary Fig. 5c). The obstacles have the same size as the active particles. The trajectory of the $i$-th particle is then obtained by solving the following Langevin equation in the overdamped regime using the second-order stochastic Runge–Kutta numerical scheme[60]

$$\dot{\mathbf{x}}_i(t) = \frac{\mathbf{F}_i(r_i, t)}{\gamma} + v_i\hat{\mathbf{u}}_i(t), \tag{4}$$

where $\mathbf{x}_i(t)$ and $\hat{\mathbf{u}}_i(t)$ are, respectively, the active particle's position and direction of motion at time $t$, $v_i$ is its speed and $\gamma$ is its friction coefficient in water. The direction of the particle's self-propulsion is defined by the unitary vector $\hat{\mathbf{u}}_i(t) = [\cos(\theta_i(t)), \sin(\theta_i(t))]$, where $\theta_i(t)$ is the particle's rotational degree of freedom given by

$$\dot{\theta}_i(t) = \Omega_i(r_i, t) + \sqrt{\frac{2}{\tau_{rot}}}\xi_i(t), \tag{5}$$

where $\Omega_i$ and $\tau_{rot}$ are the active particle's angular speed and rotational diffusion time, respectively, and $\xi_i$ is a white noise process[61]. For simplicity, we describe the cell–obstacle interaction as a superposition of three contributions: a repulsive interaction, forward scattering and random reorientations upon tumble-collision (Figs. 5a, 6, 7b, c and 8 and Supplementary Figs. 6a, 8 and 9). We modelled the first by introducing a repulsive force $\mathbf{F}_i(r_i, t)$ in the equation of motion. This force depends on the particle's distance $r_i$ from the nearest obstacle as

$$\mathbf{F}_i(r_i) = \frac{e^{-r_i}}{|r_i - d|}\hat{\mathbf{r}}_i, \tag{6}$$

where $\hat{\mathbf{r}}_i$ is the unitary vector in the direction connecting the centres of the particle and the closest obstacle. This function was chosen to reproduce a strong (local) repulsive interaction between particle and obstacle, i.e. to mimic a hardcore potential. The exponential term ensures that the force does not increase too abruptly when approaching the obstacle. To model forward scattering (the second contribution), we introduced a position-dependent angular speed $\Omega_i$ given by

$$\Omega_i(r_i) = \Omega_\infty^i(1 - e^{-\frac{r_i - d}{\ell}}), \tag{7}$$

where $\Omega_\infty^i$ corresponds to the value of the particle's angular speed in the absence of obstacles and $\ell$ is a constant that sets a length scale for the interaction. Finally, any time the particle's speed drops below $v_i/100$, a uniformly generated random angle $\in [\pi/2, 3\pi/2]$ is added to $\theta_i$ to better reproduce the experimental case of tumble-collisions (the third contribution). The values for the parameters in the simulations were chosen to closely reproduce the experimental values: $d = 3\ \mu m$, $\ell = 10.7\mu m$, $v_i = 11.0\ \mu m\ s^{-1}$ and $\Omega_\infty^i = \pm 0.8$rad s$^{-1}$. In a second version of the model, only the first two contributions (the repulsive interaction and forward scattering) were considered (Fig. 5b and Supplementary Fig. 6b), while in a third version of the model only the repulsive interaction was considered (Fig. 5c and Supplementary Fig. 6c). Lastly, in a fourth version of the model both repulsion and tumble-collisions were considered (Supplementary Fig. 7). For each value of $\rho$, we simulated 30 different obstacle configurations with 100 non-interacting particles each during 300 s. The simulated data were analysed as the experimental ones.

**Calculation of the cells' average mean square displacement**. For a given value of $\rho$, the calculation of the average mean square displacement (MSD) was performed as an ensemble average according to $\mathrm{MSD}(\tau) = \langle \mathrm{MSD}_i(\tau) \rangle$, where $\mathrm{MSD}_i(\tau) = \overline{|\mathbf{x}_i(t+\tau) - \mathbf{x}_i(t)|^2}$ is the MSD of the $i$-th cell calculated from its trajectory $\mathbf{x}_i(t)$ as a time average. The MSDs from simulations were calculated from individual trajectories whose translational and angular speeds were drawn from two Gaussian distributions respectively centred at $\langle v \rangle$ and $\langle \Omega \rangle$ and with standard deviations that match the experimental ones.

**Reporting summary**. Further information on research design is available in the Nature Research Reporting Summary linked to this article.

## Data availability
Data supporting the findings of this study are available in figshare with the digital object identifier 10.6084/m9.figshare.7981976 (https://doi.org/10.6084/m9.figshare.7981976)[62]. Further data and resources in support of the findings of this study are available from the corresponding author upon reasonable request.

## Code availability
The codes that support the findings of this study are available from the corresponding authors upon reasonable request.

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

## Acknowledgements

We thank Loris Rizzello, Melisa Canales and Saga Helgadottir for initial help in perfecting the bacterial culture protocols. We are grateful to Giovanni Volpe for critical reading of the manuscript. S.M. and G.V. acknowledge support from the Wellcome Trust [204240/Z/16/Z]. V.B. and N.A. acknowledge financial support from the Portuguese Foundation for Science and Technology (FCT) under Contract nos. PTDC/FIS-MAC/28146/2017 (LISBOA-01-0145-FEDER-028146) and UID/FIS/00618/2019.

## Author contributions

G.V. conceived the idea for the work. S.M. performed the experiments. V.B. and N.A. developed the model. S.M. and G.V. analysed the data and wrote the manuscript. N.A., L.C. and G.V. supervised the project. All authors discussed the results and revised the final version of the manuscript.

## Additional information

**Competing interests:** The authors declare no competing interests.

