## [Peer Review File · Nature Communications]

REVIEWERS' COMMENTS:

Reviewer #1 (Remarks to the Author):

In this manuscript, the authors show experimentally and computationally that obstacles comparably sized to *E. coli* bacteria can enhance the directional transport of bacteria in a thin film and promote detachment. This enhancement occurs at low obstacle densities and is lost at high obstacle densities. Using simulation, the authors show that two types of cell-obstacle interactions, forward-scattering and head-on collisions, drive this rectification of motion.

The authors have done a careful and credible job in responding to the previous review comments (from Nature Physics). The use of simulation to identify fundamental physical mechanisms is a particular strength. The results will interest a broad range of scientists studying active soft matter and microbiology, among others. Because there are few systematic studies of the effects of comparably-sized obstacles on near-surface bacterial motion, I recommend acceptance after the authors address the following issues:

1. p. 3 and 9: The average translational speed is reported to too many significant digits.
2. p. 10: I do not agree that a ten-micron-thick chamber is "in the absence of quasi-2-D confinement" for the *E. coli* bacteria -- the chamber's thickness is still a small multiple of the cell body length. Simply treating the bacteria as multiple-micron-sized objects, they are still confined at 10 microns. The authors should consider restating this argument, or show that in a thick chamber the bacteria still exhibit enhanced detachment.
3. p. 11: The authors state and attempt to show, through Figure 4 + Figure S7, that forward scattering is the primary mechanism behind the non-monotonic trends in the persistence length. They do this by showing $R + FS + TC$ (non-monotonic), $R + FS$ (non-monotonic), R (monotonic). I think, to conclude that TC cannot generate the forward scattering, that they also need to examine $R + TC$.

Reviewer #2 (Remarks to the Author):

The authors clarified to the points I brought up and made the suggested changes. The manuscript now is improved and I recommend publication. I look forward to seeing it in print and the authors' follow-up work.

Enkeleida Lushi

Reviewer #3 (Remarks to the Author):

The authors have addressed the criticism and improved presentation, therefore I recommend publication of the manuscript in Nature Communications.

Manuscript NCOMMS-19-10227A: Authors' Response to Reviewers' Comments

We are grateful to the Reviewers for finding our revised manuscript suitable for publication in *Nature Communications*. Here we address their remaining points.

Reviewer #1

In this manuscript, the authors show experimentally and computationally that obstacles comparably sized to E. coli bacteria can enhance the directional transport of bacteria in a thin film and promote detachment. This enhancement occurs at low obstacle densities and is lost at high obstacle densities. Using simulation, the authors show that two types of cell-obstacle interactions, forward-scattering and head-on collisions, drive this rectification of motion.

The authors have done a careful and credible job in responding to the previous review comments (from Nature Physics). The use of simulation to identify fundamental physical mechanisms is a particular strength. The results will interest a broad range of scientists studying active soft matter and microbiology, among others. Because there are few systematic studies of the effects of comparably-sized obstacles on near-surface bacterial motion, I recommend acceptance after the authors address the following issues:

We thank the Reviewer for recognizing the interest of our results and for their recommendation for publication in *Nature Communications*. Below we have addressed the issues raised.

1. p. 3 and 9: The average translational speed is reported to too many significant digits.

We have reduced the number of significant digits for the average translational speed on both occurrences.

2. p. 10: I do not agree that a ten-micron-thick chamber is "in the absence of quasi-2-D confinement" for the E. coli bacteria -- the chamber's thickness is still a small multiple of the cell body length. Simply treating the bacteria as multiple-micron-sized objects, they are still confined at 10 microns. The authors should consider restating this argument, or show that in a thick chamber the bacteria still exhibit enhanced detachment.

We thank the Reviewer for raising this point. Following the Reviewer's suggestion, we are now referring to "thicker sample chambers" rather than "absence of quasi-2D confinement" on page 7 (second last line).

3. p. 11: The authors state and attempt to show, through Figure 4 + Figure S7, that forward scattering is the primary mechanism behind the non-monotonic trends in the persistence length. They do this by showing $R + FS + TC$ (non-monotonic), $R + FS$ (non-monotonic), R (monotonic). I think, to conclude that TC cannot generate the forward scattering, that they also need to examine $R + TC$.

We have included new simulations showing the case where only repulsion and tumble-collisions are considered as new Supplementary Fig. 7, which we now discuss on the last sentence on page 8 of the manuscript. This case also shows a monotonic trend further confirming our hypothesis in the manuscript that forward scattering is the primary mechanism leading to enhanced propagation.

Reviewer #2

The authors clarified to the points I brought up and made the suggested changes. The manuscript now is improved and I recommend publication. I look forward to seeing it in print and the authors' follow-up work.

We thank the Reviewer for their recommendation for publication.

Reviewer #3

The authors have addressed the criticism and improved presentation, therefore I recommend publication of the manuscript in Nature Communications.

We thank the Reviewer for their recommendation for publication.